# The Significance of Central Segregation of Continuously Cast Billet on Banded Microstructure and Mechanical Properties of Section Steel

**Fujian Guo [1], Xuelin Wang [1], Jingliang Wang [1], R. D. K. Misra [2] and Chengjia Shang [1,3,***

[1]  Collaborative Innovation Center of Steel Technology, University of Science and Technology Beijing, Beijing 100083, China; carterguo2013@163.com (F.G.); xuelin2076@163.com (X.W.); jlwang@ustb.edu.cn (J.W.)

[2]  Department of Metallurgical and Materials Engineering, University of Texas EI Paso, EI Paso, TX 799968, USA; dmisra2@utep.edu

[3]  State Key Laboratory of Metal Materials for Marine Equipment and Applications, Anshan 114021, China

*   Correspondence: cjshang@ustb.edu.cn; Tel.: +86-10-6233-2428

**Abstract:** The solidification structure and segregation of continuously cast billets produced by different continuous casting processes are investigated to elucidate their effect on segregated bands in hot-rolled section steel. It suggested that segregated spots are mainly observed in the equiaxed crystal zone of a billet. The solidification structure is directly related to superheating and the intensities of secondary cooling. To a certain extent, the ratio of the columnar crystal increases with the increase of superheating and secondary cooling. Moreover, the number of spot segregations decreases with the decrease of the equiaxed crystal ratio. After hot rolling, the segregation spots are deformed to form segregated bands in steels. The severe segregation of Mn in segregated bands corresponds with that in the segregation spots. The elongation ratio and low temperature toughness deteriorate significantly by a high fraction of degenerate pearlite caused by central segregation. With a decrease of central segregation, the total elongation is increased by 10% and the ductile–brittle transition temperature (DBTT) is also reduced from −10 to −40 °C. According to the experimental results, columnar crystal in billets is preferred to effectively reduce the degree of central segregation and further improve low temperature toughness and the elongation ratio.

**Keywords:** continuous casting process; solidification structure; central segregation; banded microstructure; mechanical properties

## 1. Introduction

With the increasing application of section steel, the service environment is becoming harsher and the requirements for low temperature toughness are getting higher, e.g., in arctic regions. Compared to plates, the rolling reduction and, consequently, deformation accumulation of section steels are relatively low, which lead to insufficient recrystallization and coarse prior austenite grains [1]. Moreover, the microstructures of ferrite and perlite are difficult to refine due to the lack of controlled cooling process [2,3]. Therefore, low temperature toughness is always a challenge for section steel [4]. For high strength low alloy section steel, besides grain refinement [5], a banded microstructure is considered to be an important factor that affects mechanical properties [6]. A banded microstructure is always related to the cooling rate of ferrite/perlite phase transformation [7]. However, the central segregation of C, Mn, and other elements during solidification also result in the formation of kinds of abnormal band microstructures, like pearlite [8], bainite [9] and martensite [10,11].

The quality of a continuously cast billet is closely related to the continuous casting processes, such as superheating, secondary cooling [12], electromagnetic stirring [13] and soft reduction [14].

For large size billets, the continuous casting process directly affects their solidification structures (equiaxed/columnar crystal ratio). Due to the influence of various factors, such as selective crystallization and the solidification structure, the segregation of C, Mn and other alloys in the billet are inevitable [15, 16]. Previous studies have shown [10,11] that central segregation in a billet drastically deteriorate the billet's mechanical properties—low temperature toughness, in particular. Meanwhile, a banded microstructure is also an internal defect of steel that deteriorates mechanical properties. Though the relationship between central segregation and continuous casting parameters has been reported in previous studies [12–14], the heredity effect of central segregation on segregated bands and mechanical properties is still a problem that has not yet been solved. A few works have been done to examine the relative influences of central segregation of billets on the banded microstructures and mechanical properties of final products. Thus it is more important to study the influence of the central segregation of billets on banded microstructures than the rolling and cooling process [17].

In this paper, the macrostructure and microstructure of billets and steels under two continuous casting process conditions are studied; the relationship between central segregation, banded structure and the mechanical properties are analyzed; and the optimization method of billet quality is discussed.

## 2. Materials and Methods

The materials used in this study were U-section steels, which were obtained from the continuously cast billet with a cross-section of $280 \times 380$ mm$^2$. The composition of the steel in weight percent (wt%) was 0.15C–0.28Si–1.5Mn–0.07V–0.01N. The design goal of this experiment was to study the effect of solidification structures on the degree of segregation, homogeneity of microstructure, and low temperature toughness. As such, just two set of parameters were provided, and the relevant parameters of the continuous casting process are shown in Table 1 (high superheating and high secondary cooling intensity led to a low proportion of equiaxed structure; stop using EMS (electro magnetic stirring) promoted columnar crystal growth and enlarged the columnar crystal region). Two corresponding billets are referred to as Billet 1 and Billet 2. After rolling, the corresponding samples were renamed Steel 1 and Steel 2. The billet samples were cut after continuous casting under a stable state, and the continuously cast billet was rolled by using the same rolling process that excluded influence by rolling factors.

**Table 1.** The parameters of the continuous casting process of the experimental steel.

| Billet | Superheating/°C | Speed/m/min | Secondary Cooling Kg/L | EMS |
|--------|-----------------|-------------|------------------------|---------|
| 1 | 24 | 0.7 | 0.25 | with |
| 2 | 35 | 0.7 | 0.4 | without |

The billet and steel samples were taken from the head of the experimental billet and the corresponding U-section steel. The continuously cast billet was evaluated in terms of macrostructure, and its subsequent effect on the mechanical properties and microstructure of the U-section steel was studied. The sampling region for the billets and the U-section steels from the neighboring region was selected to study the severity and distribution of segregation. The central segregation of the billet and the U-section steel were revealed by 50 pct HCl at 80 °C for 0.5 h.

Samples for microstructure observation were cut from the central area of the black rectangle on the steel and then mounted and mechanically polished by using a standard metallographic procedure. The specimens were etched with 3% nital for optical microscopy (OM) and scanning electron microscopy (SEM) observations. Segregation behavior was analyzed by electron probe micro-analysis (EPMA, JXA-8530F, Tokyo, Japan). Prior to EPMA analysis, the segregation bands were revealed with a solution that contained picric acid, detergent, carbon tetrachloride, and sodium chloride. The etching temperature was ~60 °C, and the soaking time was 2 min. The CCT diagram was calculated with the JMatPro 7.0 according to the alloy content that was measured by EPMA in segregation area.

The temperature in the center and quarter positions of the billet was calculated by the Procast software (version 7.5, ESI Group, Paris, France) according to the continuous casting parameters shown in Table 1.

CVN impact tests were conducted by using full-size Charpy specimens with dimensions of $55 \times 10 \times 10$ mm that were machined according to the ISO 148-1 standard from the geometrical center of the black rectangle on the steel along the longitudinal direction. CVN impact tests were conducted at temperatures of 0, −20 and −40 °C. Dog bone-shaped round tensile specimens (gage length: 110 mm; diameter: 8 mm; and orientation: longitudinal) were prepared, and the test was conducted at room temperature. The Vickers micro-hardness of the segregation band was measured by using a load of 1 Kg and a dwell time of 15 s. The load was selected to ensure that the indent could simultaneously overlap both ferrite and pearlite.

## 3. Results

### 3.1. Macro-Etching Results of Billets and U-Section Steels

The macro-etching images of Billets 1 and 2 are shown in Figure 1a,b, respectively. The obvious difference was present in the solidification structure. The equiaxed crystal ratio of Billet 1 was higher (19.1%), and the small segregation spots (black spots) were dispersed in the equiaxed crystal region; the equiaxed crystal ratio of Billet 2 was only 2.1%, and several shrinkage holes were observed in the equiaxed crystal region. Figure 1c,d shows a magnified view of the center region of Figure 1a,b, where the characteristics can be seen more clearly. Many black spots were distributed in the equiaxed crystal area of Billet 1, while the segregated spot in the equiaxed crystal area of Billet 2 become smaller, with the exception of the presence of several shrinkage holes.

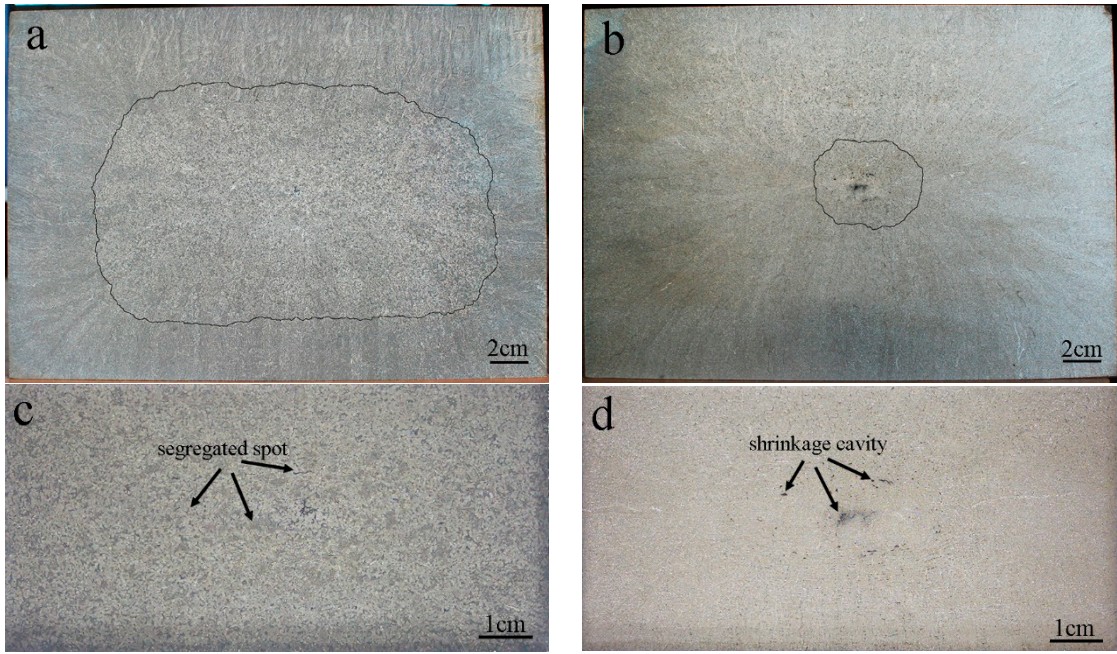

**Figure 1.** *Cont.*

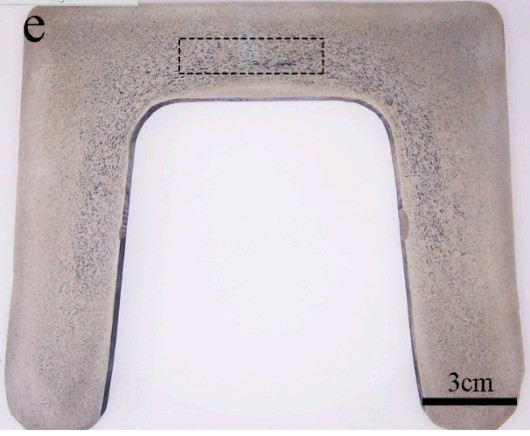
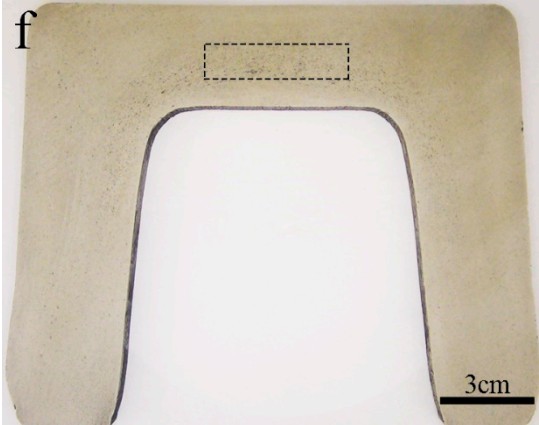

**Figure 1.** Macro-etching results of continuously cast billet and U-section steel. (**a**,**b**) Billets 1 and 2; (**c**,**d**) the enlarged image of the center region of Billets 1 and 2; (**e**,**f**) Steels 1 and 2. (Samples for microstructure observation and mechanical properties were cut from the central area of the black rectangle on the steel, shown in Figure 1**e**,**f**).

After hot rolling, the segregation spots in the equiaxed crystal region were compressed to the inner arc of U-section steel, as shown in Figure 1e,f. Dense segregation spots were severely present in the cross-section of Steel 1, while just a small number of segregated spots were observed in Steel 2, and the shrinkage holes disappeared. Thus, the segregation region could not be eliminated in the subsequent process, but the shrinkage holes could be welded during the rolling process.

*3.2. Enrichment of Elements in Continuous Casting Billets and U-Section Steel*

The segregating spots in the equiaxed crystal region of the billet were analyzed by EPMA. The test samples were cut from an identical position in the central area of the billets. The mapping images of the segregating elements C and Mn are depicted in Figure 2, which suggests the black spots in the equiaxed crystal region were enriched by C and Mn and the degree of segregation of Billet 1 (Figure 2b,c) was higher than that of Billet 2 (Figure 2e,f).

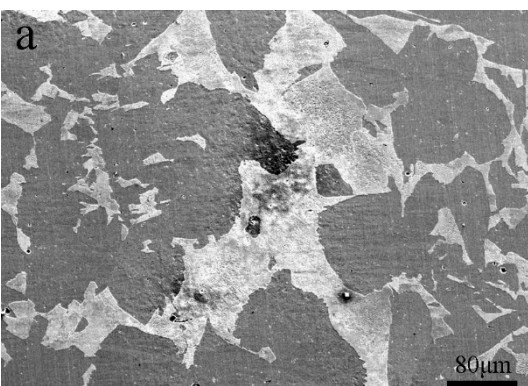
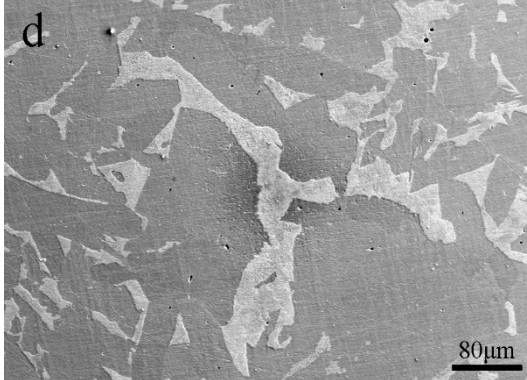

**Figure 2.** *Cont.*

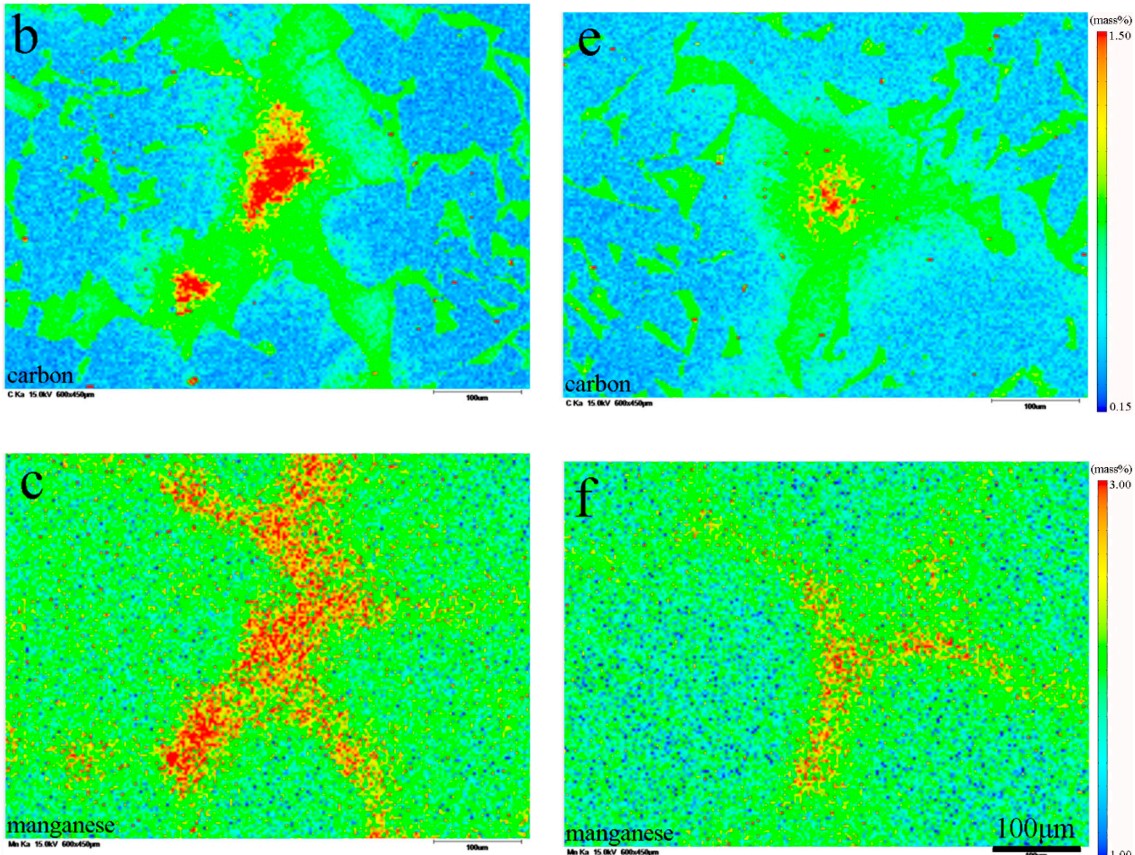

**Figure 2.** Two-dimensional mapping of solute elements in the billet sample: (**a**–**c**) Billet 1; (**d**–**f**) Billet 2.

After hot rolling, the segregating spots in the billet were compressed to the segregating band. In order to investigate the distribution of the enriched elements in the microstructure, the composition of the microstructure was measured by the map scanning of EPMA. The segregated band was etched by picric acid, as shown in Figure 3a,d. The black bands were identified as segregation bands, which were inherited from the central segregation in the equiaxed crystal region of the billets. The number of segregation bands in Steel 1 was larger than that of Steel 2, with the width of segregation band in Steel 1 being larger than that of Steel 2. The carbon segregation (Figure 3b,e) in the microstructure showed that carbon was enriched in pearlite, while Figure 3c,f shows that the black bands were caused by Mn enrichment. The above results suggest that the degree of segregation in Steel 1 was more severe than that of Steel 2 and was consistent with the degree of central segregation in the billets.

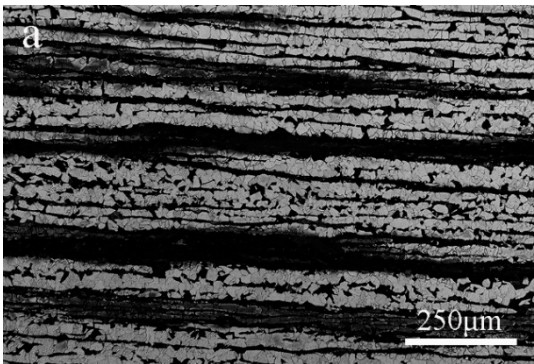
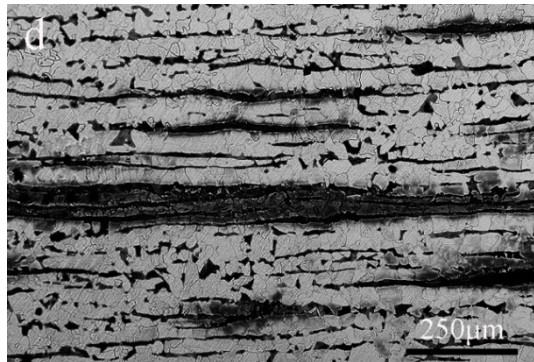

**Figure 3.** *Cont.*

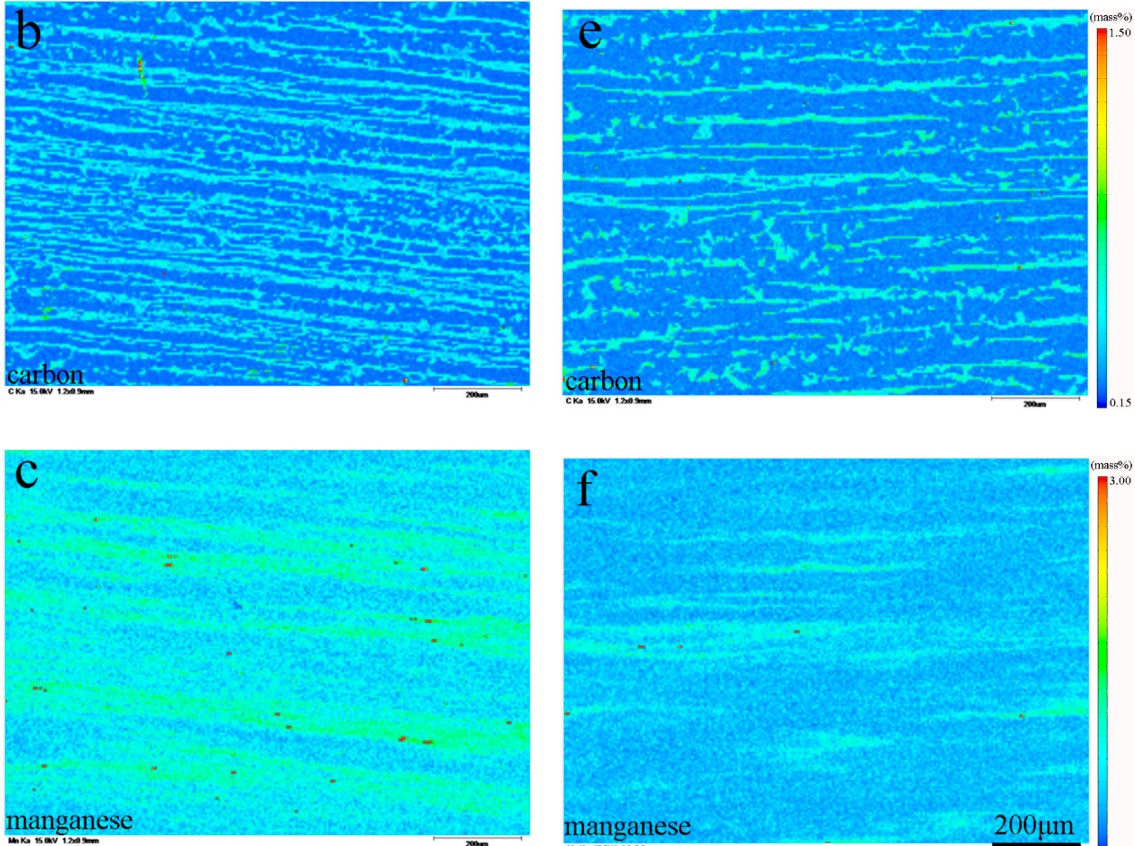

**Figure 3.** Segregation studies by electron probe micro-analysis (EPMA) mapping in the segregation region of steel: (**a–c**) Steel 1; (**d–f**) Steel 2.

EPMA line scanning was used to further study the segregation behavior of carbon and manganese, and the results are shown in Figure 4. The microstructures obtained in Figure 4a,b are pearlite and ferrite and are displayed as gray and dark morphologies, respectively. Line scanning results (Figure 4c,d) indicated that the peak of carbon enrichment emerged at the region of pearlite. Though there was no difference in the carbon enrichment of both samples, the peak and fluctuation of Mn enrichment suggested that the degree of Mn segregation in Sample 1 was significantly larger than that of Sample 2. The mean contents of Mn in Samples 1 and 2 were 1.98% and 1.81%, respectively.

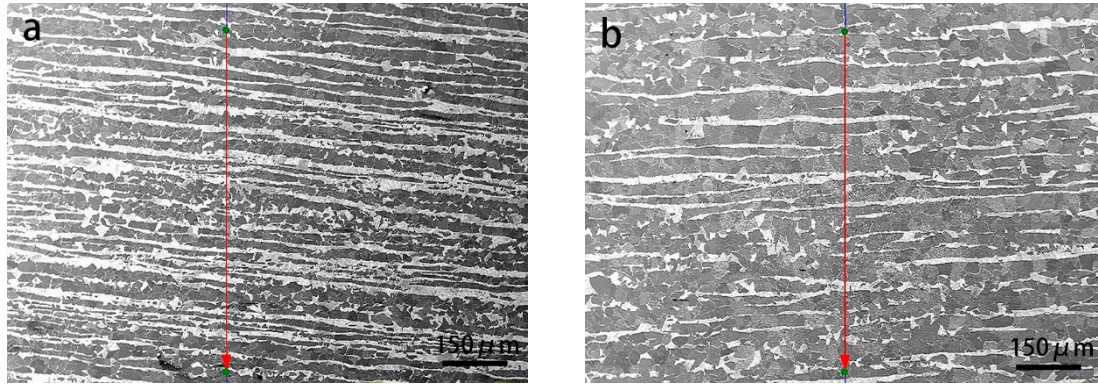

**Figure 4.** *Cont.*

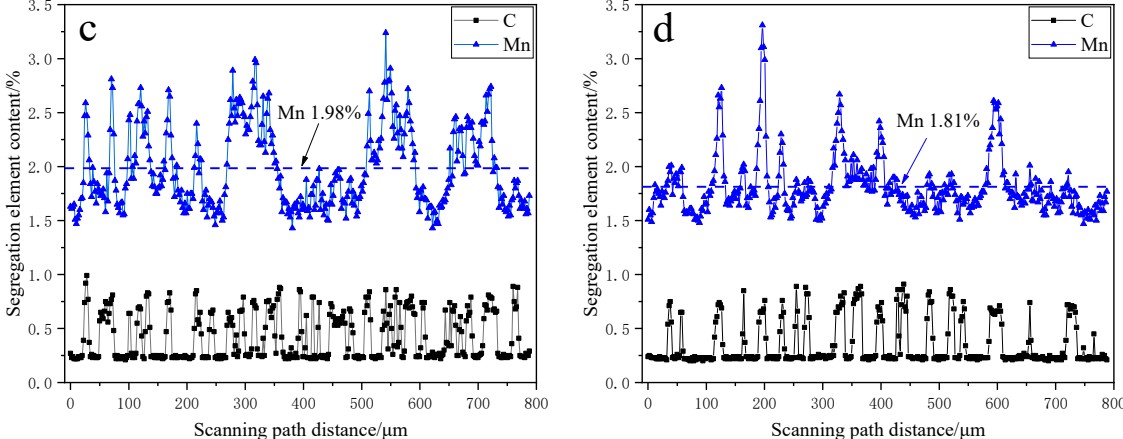

**Figure 4.** Segregation element content measured by EPMA line scanning in the segregation region of steel: (**a**,**c**) Steel 1; (**b**,**d**) Steel 2.

### 3.3. Microstructure in U-Section Steel

Samples for microstructure observation were cut from the center of the black rectangle (Figure 1e,f) on a cross-section. The microstructure of this steel is shown in Figure 5. It consisted of ferrite and pearlite in the black rectangle region (Figure 1e,f). However, the difference in microstructure was obvious. The fraction of pearlite in Steel 1 (Figure 5a,c) was high (41%), while the fraction of pearlite in Steel 2 (Figure 5b,d) was just 14%. The morphogenesis of pearlite is shown clearer in a magnification image (Figure 5e–h). The microstructures in the rectangular areas of Figure 5c,d are zoomed in Figure 5e,f. The pearlite in the two steels was different, as the spherical and discontinuous pearlite (Figure 5e) was degenerate pearlite in Steel 1 while the pearlite in Steel 2 was uniformly aligned as lamellar (Figure 5f). In addition, Figure 5e,f are further zoomed in to observe the morphology of different pearlite structures, as shown in Figure 5g,h.

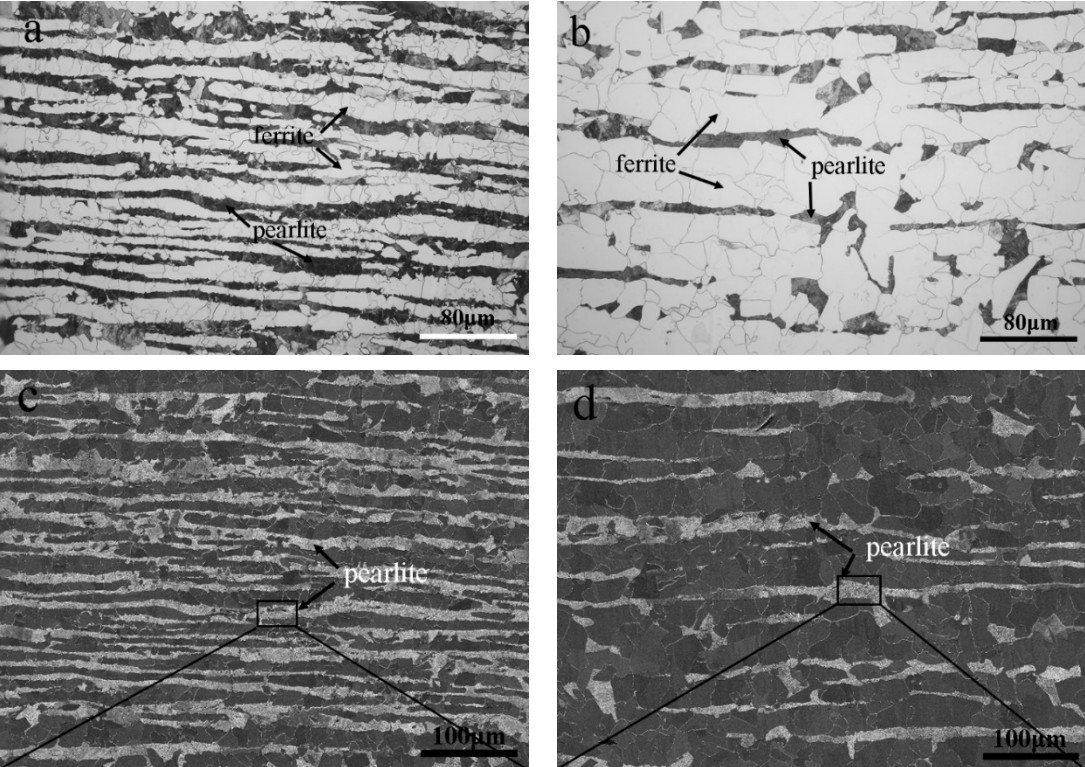

**Figure 5.** *Cont.*

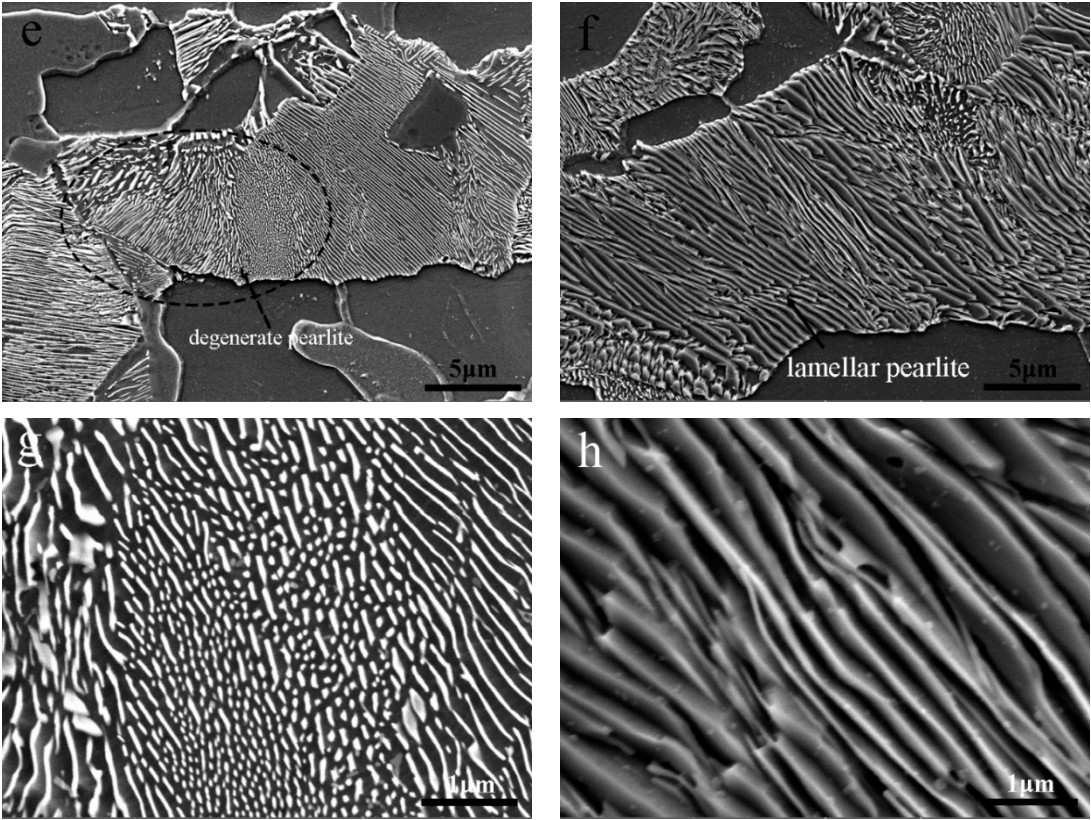

**Figure 5.** The microstructures of the U-section steel seen by an optical microscope (**a**,**b**) and scanning electron microscope (**c**–**h**). (**a**,**c**,**e**,**g**) Steel 1; (**b**,**d**,**f**,**h**) Steel 2.

*3.4. Mechanical Properties*

The tensile strength (TS), yield strength (YS), uniform elongation (Agt), and total elongation (A) of the test specimens were measured, and the results are shown in Table 2. By comparing the mechanical properties of Steel 1 and Steel 2, it may be noted that yield strength of Steel 1 was higher than that of Steel 2 by ~110 MPa, and the tensile strength of Steel 1 was higher than that of Steel 2 by ~20 MPa. The uniform elongation of Steel 1 was lower than that of Steel 2 by ~6.5%, and the total elongation of Steel 1 was lower than that of Steel 2 by ~10%. The above results suggest that the yield strength of Steel 1 was significantly higher than that of Steel 2, and the elongation of Steel 1 was obviously lower than that of Steel 2. The microstructure of the U-section steel was composed of ferrite and pearlite bands. The microhardness of the segregation area was measured at the geometrical center of the black rectangle (Figure 1e,f) in the U-section steel. A total of 10 measurements were taken to obtain an average value. In order to ensure the microhardness accuracy, 1 Kg was selected for the overlapping of both ferrite and pearlite. The microhardness values of Steels 1 and 2 were ~190 and 162 HV, respectively, as shown in Figure 6a. It can be concluded that the microhardness of Steel 1 was higher than Steel 2, as could be seen from the tensile strength and the corresponding microstructures.

**Table 2.** Tensile properties of experimental steel.

| Steel | YS/MPa | TS/MPa | Agt/% | A/% |
|---|---|---|---|---|
| 1 | 504.5 (±9.2) | 572.0 (±4.2) | 8.53 (±1.05) | 25.36 (±0.79) |
| 2 | 386.0 (±2.8) | 552.5 (±0.7) | 15.04 (±0.30) | 34.40 (±0.10) |

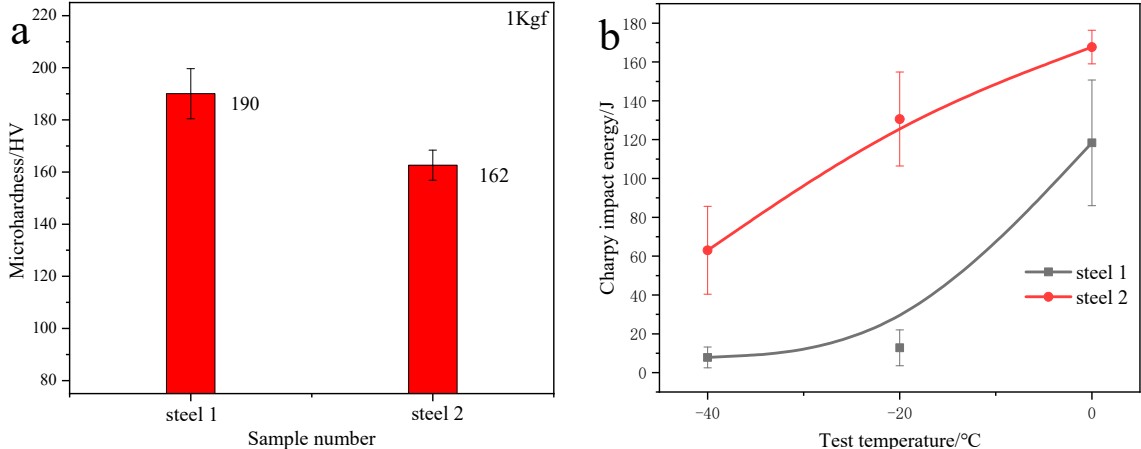

**Figure 6.** Microhardness (**a**) and Charpy impact toughness (**b**) of the U-section steel.

As U-section structural steels were used in a low temperature environment, the superior low temperature toughness had to satisfy the target requirements. The ductile–brittle transition temperature (DBTT) of the two steels was measured, and the experimental results are shown in Figure 6b. It was clear that Steel 2 showed an excellent low temperature toughness with a DBTT of −40 °C; the toughness of Steel 1 did not meet the requirements of the low temperature, as its DBTT was −10 °C.

## 4. Discussion

### 4.1. Occurrence of Degenerate Pearlite and Its Influence on Mechanical Properties

The central segregation in the continuously cast billet could not be eliminated by subsequent processing and was directly inherited in the subsequently rolled steel, and this issue was confirmed in the macro-etched results (Figure 1e,f) of the rolled steel. When the central region of the continuously cast billet had severe segregation (Figure 1a), Mn segregation was also inherited in the microstructure of the rolled steel (Figures 2 and 3). Previous studies [9,18–20] have suggested that the segregation of solute elements in a slab or billet affects the transition temperature (Ar3) of austenite to ferrite during the cooling process and becomes the source of banded segregation. The CCT diagram was calculated by the JMatPro software according to the alloy content that was measured by EPMA in the segregation area (Steel 1, 0.15C–1.98Mn–0.28Si–0.07V–0.01N; Steel 2, 0.15C–1.81Mn–0.28Si–0.07V–0.01N). As shown in Figure 7, the phase transformation temperature of Steel 1 was obviously lower than that of Steel 2. Mn is an austenite stabilizing element and can reduce Ar3, so the Ar3 in a rich-Mn region is lower than that in a poor-Mn region [21]. As temperature decreases, proeutectoid ferrite preferentially nucleates in the poor-Mn dendrites. As ferrite grains grow during the longitudinal extension of the deformation process, adjacent ferrites form bands during the rolling process. In addition, since the solubility of C in ferrite is much lower than that of austenite, with the formation and growth of ferrite, C is continuously rejected to the area around austenite that has not been transformed. At the same time, it is known that the concentration of the Mn element in this region is high, and Mn will reduce the activity of C and hinder its diffusion [9]. Therefore, in the subsequent transformation process, the austenite in the segregation region easily forms pearlite, and, here, the proportion of pearlite in Steel 1 was higher than that of Steel 2 (Figure 5). In addition, the formation of degenerate pearlite was caused by the enrichment of C and Mn in the segregated region. During continuous cooling phase transformation, the degenerated pearlite is transformed from carbon-rich austenite. In general, during the phase transformation from austenite to ferrite, carbon is enriched to untransformed austenite. Carbon-rich austenite has a higher stability and a lower transformation temperature. At lower transformation temperatures, the transformation driving force is larger, resulting in the separate formation of cementite and ferrite. Finally, the morphology of pearlite is spherical or discontinuously lamellar. The morphology of pearlite has also been reported in the literature [22], where it has been

suggested that at high reaction temperatures, group nodules of pearlite form that consist of many colonies of pearlite; at lower reaction temperatures, pearlite nodules form as hemispheres or as sectors of spheres.

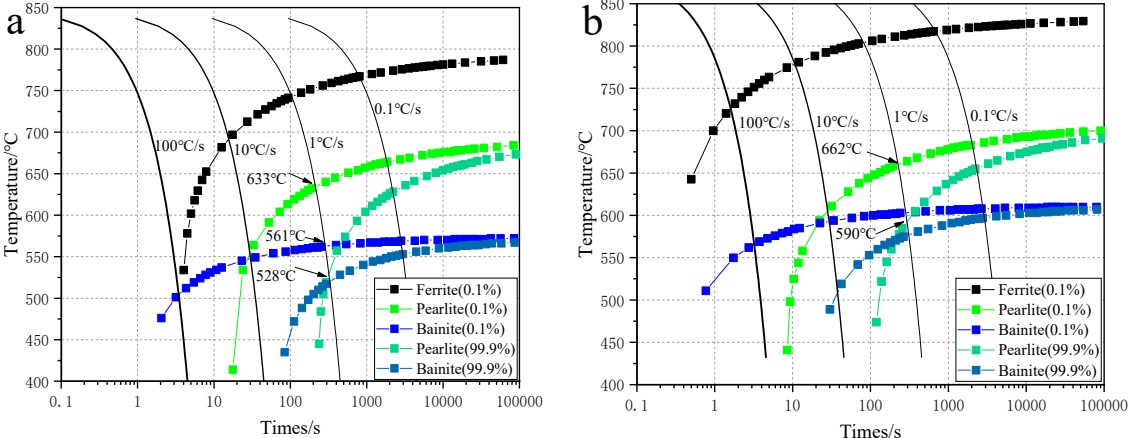

**Figure 7.** CCT diagram of the central segregation region of steel. (**a**) Steel 1; (**b**) Steel 2.

Mechanical properties are caused by the microstructure of material. In our experiments, the elongation and toughness of pearlite was lower than that of ferrite, and the strength and hardness of pearlite was higher than that of ferrite. In addition, the hardness of the degenerate pearlite was higher than the lamellar pearlite. The degenerate pearlite did not have good plastic deformation. Therefore, the strength of Steel 1 was significantly higher than that of Steel 2. On the other hand, when an external load was applied, stress concentration tended to occur at the interface between the degenerate pearlite and ferrite. The more severe the stress concentration, the easier it was for the crack to nucleate and propagate. Thus, the low temperature toughness of Steel 2 was obviously higher than that of Steel 1.

### 4.2. Effect of Continuous Casting Process on Solidification Structure and Central Segregation of Billet

In our present study of low alloy steels, as the superheating and secondary cooling intensity increased, the area affected by the central segregation was reduced (Figure 1a,b), and the segregation spots in the equiaxed crystal zone significantly decreased (Figure 1c,d). The result was consistent with Ji's study [21], where segregating spots decreased with the decreased round bloom of the equiaxed crystal zone. Previous studies [23] have suggested that factors affecting solidification structures include high superheating temperatures that lead to a high proportion of columnar structures, stop using EMS (electro magnetic stirring) that promotes columnar crystal growth and enlarges the columnar crystal region, and a high secondary cooling intensity that increases the columnar crystal ratio. In addition, according to the model given by Equation (1) [24] of the columnar-equiaxed transformation, the higher the secondary cooling intensity and superheating temperature, the larger temperature gradient; meanwhile, the other values remain unchanged, and the ratio of $G_n$/V more easily exceeds the critical value of the columnar crystal region, such the columnar crystal region proportion is enlarged. Thus, two kinds of solidification structures were obtained in Billets 1 and 2, and their equiaxed crystal ratios were ~19.1% and 2.1%, respectively.

$$G_n/V > C_{st} = a(8.6\Delta T_0 \frac{N_0^{\frac{1}{3}}}{n+1})^n \tag{1}$$

where $G_n$ is the temperature gradient, V is the grain growth rate, $a$ is the alloy material constant, $\Delta T_0$ is the superheating value, $N_0$ is the nucleation density, and $n$ is the alloy material related constant.

Central segregation is the result of the selective crystallization of solute elements in the solidification front during the continuous casting process. The solubility of solute elements in the solid phase

is lower than that of the liquid phase. The diffusion of solute elements during solidification and selective crystallization is directly related to the cooling rate during the continuous casting process. The cooling rate in solidification provides kinetic and thermodynamic conditions for segregation behavior. The temperature distribution in the center and quarter positions of the billet was calculated by the Procast software according to the continuous casting parameters in Table 1. The temperature profile in the center and quarter positions of the billet is shown in Figure 8. The cooling rate of Billet 1 is lower than that of Billet 2. That is, when the cooling rate was low, the solute element precipitated in the solidification front and dissolved into the liquid phase, eventually forming the central segregation region in the center of the continuously cast billet; on the contrary, a small amount or no solute element diffused at the solidification front from the solid to liquid phases at a high cooling rate, and the central segregation of the continuously cast billet was suppressed. In addition, the segregation spots were formed due to the segregation of elements that were transported to the small cells or the free spaces in the mushy zone [8]. The center of the equiaxed crystal zone may have provided an opportunity to increase the volume of the small cells, which presented the black spots in the equiaxed crystal zone. Therefore, in the continuously cast billet of the low alloy steel, the formation of central segregation was suppressed by the decrease of the equiaxed crystal zone.

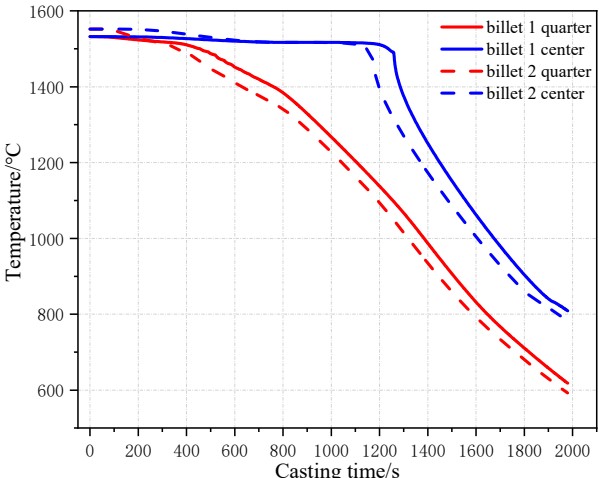

**Figure 8.** The temperature of the quarter and center positions of the billet in the solidification process.

## 5. Conclusions

A comparative study of U-section steels rolled from continuously cast billets with different internal qualities showed that solidification structure had a significant effect on the degree of central segregation in billets, and the segregation affected these billets' microstructures, distributions of solute elements and mechanical properties. The conclusions are as follows:

(1) The central crystal zone of a continuously cast billet presented segregated spots that were caused by the concentration of Mn.

(2) With the application of high superheating and secondary cooling temperatures without EMS (electro magnetic stirring), the fraction of the equiaxed crystal zone effectively decreased. The columnar crystal structure was more effective to control central segregation in the continuously casting billet.

(3) The microstructure of the steel that was rolled from the billet with severe segregation consisted of ferrite and degenerate pearlite, while the microstructure of the steel that was rolled from the optimized billet was dominated by ferrite and lamellar pearlite. The fraction of pearlite in Steel 1 (41%) was obviously higher than that of Steel 2 (14%).

(4) Controlling the degree of central segregation could achieve an excellent low temperature toughness, as the DBTT decreased from −10 to −40 °C, and the elongation increased by ~10%.

**Author Contributions:** C.S., R.D.K.M., J.W. and X.W. designed and supervised the research; F.G. performed the experiments, analyzed the data, and wrote the paper. All authors have read and agreed to the published version of the manuscript.

**Funding:** This work is financially supported by National Key Research and Development Project of China (2017YFB0304900 and 2017YFB0304700), Anshan Iron and Steel Group Company, State Key Laboratory of Metal Materials for Marine Equipment and Applications.

**Conflicts of Interest:** The authors declare no conflict of interest.

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
