# Peer review of "The Significance of Central Segregation of Continuously Cast Billet on Banded Microstructure and Mechanical Properties of Section Steel"

_metals, doi:10.3390/met10010076_

Round 1

Reviewer 1 Report

I have read through the manuscript which outlines a manufacturing experiment whereby two billets were continuously cast under different conditions and then rolled to form U-shaped continuous structures. Columnar to Equiaxed Transitions were observed, which were also shown to coincide with segregation patterns of Carbon and Manganese.

After rolling, the banded structures were observed and were assumed to be dependent on the initial grain structure.

A series of compositional and mechanical tests were performed and various conclusions have drawn about the results. The conclusions seem reasonable and appear to be supported by the investigative analyses. 

A few corrections are suggested:

Overall, please proof read the document for English grammar.

Please write out the full acronym for EMS (Electro Magnetic Stirring).

In table 1 please clarify the units of secondary cooling as Kg/L seems unusual.

The authors state that the billets were "rolled using the same rolling process, which excludes the influence of rolling factors". Please give these parameters if known.

In the discussion section, please expand on the details of how the two graphs in figure 7 were generated. Please provide the input parameters used in the JMAtPro.

Also, please expand the discussion to include more comments on figure 8. Explain how was this graph generated. Was it through measurement or simulation? This is an important point.

Author Response

Dear reviewer

We would like to thank the reviewer for careful and thorough reading of this manuscript and for the thoughtful comments and constructive suggestions, which help to improve the quality of this manuscript. The reply was submitted in the attachment. Please find it. Thank you!

Best regards

Author 

Reviewer 2 Report

The paper provides an analysis of the solidification and segregation processes in continuously cast billets, which are produced by means of different continuous casting processes.

The topic of the paper is interesting and is in line with the aims and scope of the journal. However, the paper shows some major flaws, which prevent its publication in the present form.

The state of the art analysis, which is provided in the first section (Introduction), is quite superficial: most of the provided references are quite outdated and not described in details. Lump citations should be avoided as much as possible.

The experimental procedure appears rigorous and is well described. However, the authors do not provide the rationale for the selection of the used material. Moreover the number of samples is too limited to substantiate the general validity of the proposed results. These two aspects represent two major weaknesses for the overall scientific value of the paper and its interest for the readers, as they impact the generality of the achieved conclusions.

The description of the results provided in Section 3 is accurate and detailed. However, the quality of Figure 2 needs to be improved.

The discussion of the results is interesting, although the proposed outcomes are strongly affected by the above-mentioned weaknesses. In effects, in variability of the continuous casting process is not fully explored within the paper, due to the limited experimental basis (in practice, two set of parameters).

The English language absolutely needs to be revised, as there are numerous errors and unclear sentences. For instance, considering only the abstract:

Row 15: please replace “billet” with “billets”; Row 16: Please replace “process” with “processes”; Row 17-8: The sentence: “It suggested that segregated spots mainly present in equiaxed crystal zone of billet” is unclear, probably one verb is missing; Row 20: please replace “become less” with “decreases”.

In all the other sections there are numerous linguistic inconsistencies. Therefore I suggest a proofreading process from professional reviewers.

Author Response

(The authors gave the same response as above.)

Round 2

Reviewer 2 Report

The paper provides an analysis of the solidification and segregation processes in continuously cast billets, which are produced by means of different continuous casting processes.

The topic of the paper is interesting and is in line with the aims and scope of the journal.

The authors significantly improved both the analysis of the state of the art and the English language.

The explanations provided by the authors in the reply to my reviews helped me to understand how they select the material and the number of samples. I suggest that these explanations and clarifications are added also in the paper.

The quality of Figure 2 are affected by the limited dimension of each subfigure, which make the scale difficult to read. This drawback could be solved by increasing the dimension of the sub-figures and splitting the figure in two or three figures.
